# Strategies for Facilitating Totally Percutaneous Transfemoral TAVR Procedures

**DOI:** 10.3390/jcm11082104

**Published:** 2022-04-09

**Authors:** Amnon Eitan, Hussein Sliman, Avinoam Shiran, Ronen Jaffe

**Affiliations:** Department of Cardiology, Carmel Medical Center, Haifa 3436212, Israel; eitancardio@gmail.com (A.E.); dr.hussein.sliman@gmail.com (H.S.); shiran_avinoam@clalit.org.il (A.S.)

**Keywords:** transcatheter aortic valve replacement, percutaneous, transfemoral

## Abstract

Transcatheter aortic valve replacement (TAVR) has transformed the treatment of aortic stenosis and should ideally be performed as a totally percutaneous procedure via the transfemoral (TF) approach. Peripheral vascular disease may impede valve delivery, and vascular access site complications are associated with adverse clinical outcome and increased mortality. We review strategies aimed to facilitate TF valve delivery in patients with hostile vascular anatomy and achieve percutaneous management of vascular complications.

## 1. Introduction

Minimally-invasive transcatheter aortic valve replacement (TAVR) has supplanted surgical aortic valve replacement (SAVR) as the preferred mode of treatment in the majority of patients with severe aortic stenosis [1,2]. The transfemoral (TF) approach is the preferred route for valve delivery as it is less invasive than alternative vascular access sites and enables the procedure to be performed under conscious sedation [3,4]. Performing TAVR as a totally percutaneous procedure may potentially minimize vascular complications (VC), facilitate patient ambulation and shorten hospital stay [5]. Comprehensive analysis of the vascular anatomy by CT angiography is a key component of procedural planning [6]. Puncture of the common femoral artery (CFA) has traditionally been guided by fluoroscopic imaging of the femoral head and palpation of the femoral pulse. Ultrasound guidance to locate the appropriate site for vessel puncture has been reported to be beneficial in some observational studies [7,8]. Valve delivery requires insertion of a large-bore sheath into the CFA. Commonly used valve delivery systems require a minimal arterial diameter ranging between 5–6 mm (Table 1). 

Failure of vascular closure devices (VCD) to achieve access site hemostasis following valve implantation and sheath removal may lead to severe bleeding, as well as limb ischemia [9]. Percutaneous management of vascular complications may preclude the need for invasive surgical vascular repair, facilitate rapid mobilization of patients and shorten hospital stay. Analysis of a database of 45,884 TAVR procedures found that bleeding, but not VC, was an independent predictor of mortality, suggesting that prompt achievement of hemostasis may improve outcome in these patients [10]. 

Calcified, stenosed and tortuous ileofemoral arteries characterize patients with “hostile” vascular anatomy and may impede valve delivery and increase susceptibility to vascular injury during TAVR. In such patients, SAVR may potentially enable valve replacement without the need for instrumentation of the peripheral vasculature. However, patients with peripheral vascular disease often have co-morbidities which are associated with increased surgical risk and may therefore benefit from percutaneous valve delivery. Alternative vascular access approaches (e.g., transapical, transaxillary, transcaval) were developed in order to enable TAVR in patients with hostile vascular anatomy in whom TF valve delivery was considered technically challenging. No randomized trials have compared these different approaches; however, the TF route is considered the least invasive. The aim of the present review is to summarize novel techniques which may facilitate totally percutaneous TF valve delivery.

## 2. Vascular Complications Following TAVR

Peripheral vascular disease (PVD) is common in patients referred for TAVR [11,12], and is associated with increased risk for VC and mortality [13,14,15]. Vascular access site complications following TAVR include bleeding, as well as limb ischemia [16,17]. Ileofemoral arterial perforation may be caused by vascular trauma during valve delivery, as well as during sheath removal. Failure of VCD to achieve hemostasis may lead to bleeding; however, deployment of these devices may also result in femoral artery stenosis or occlusion. In a cohort of 34,893 patients who underwent TAVR between 2011 and 2016 and were included in the Society of Thoracic Surgeons/American College of Cardiology Transcatheter Valve Therapy registry, VC occurred in 9.3% of the subjects, and were associated with increased mortality and rehospitalization [18]. In a cohort of patients who underwent TAVR in 2017, only major VC (2.9% of the subjects) but not minor complications (23.2% of the subjects) were associated with increased mortality, and required surgical vascular repair in half the cases [19]. In various studies, predictors of VC included female gender, diagnosis of PVD, smaller diameter of iliofemoral arteries, vascular calcification and tortuosity, larger sheath size and increased sheath to femoral artery ratio [18,19,20,21,22]. 

Vascular calcification [23,24,25], tortuosity [26] and small vessel diameter [20] may preclude TF valve delivery. Use of stiff 0.035″ guide-wires (e.g., Lunderquist Extra-Stiff, Cook, Bloomington, IN, USA) may facilitate sheath insertion and valve delivery. In some cases, balloon angioplasty may be performed to dilate obstructive ileofemoral stenoses; however, this technique may be ineffective in the presence of severe circumferential calcification and may cause vascular injury. Dedicated devices for treating calcified vascular lesions, such as orbital atherectomy (Diamondback, CSI, St. Paul, MN, USA) and intravascular lithotripsy (Shockwave Medical, Santa Clara, CA, USA), may facilitate valve delivery in patients with occlusive peripheral vascular disease [27,28,29]. 

## 3. Vascular Closure Device Failure

Use of VCD to achieve hemostasis decreases wound complications and duration of hospitalization compared to surgical cut-down [9,30,31,32,33]. Suture-mediated VCD (Prostar XL and Proglide, Abbott Vascular, Santa Clara, CA, USA) are deployed prior to sheath insertion (pre-closure) and the sutures are fastened following sheath removal. Optimal depth for deployment of collagen plug-based VCD (Manta, Teleflex, Morrisville, NC, USA) is assessed prior to sheath insertion, and the VCD is deployed following sheath removal. Vascular closure devices may fail to achieve access site hemostasis in two distinct settings: inability to position a VCD at the beginning of the procedure (“primary” VCD failure), and inability of a VCD to achieve hemostasis following sheath removal at the end of the procedure (“secondary” VCD failure). In cases of primary VCD failure, inability to perform pre-closure may lead to abortion of the procedure, surgical femoral artery cut-down or selection of more invasive alternative access, and may be associated with worse clinical outcome [34]. Secondary VCD failure, with the inability to achieve hemostasis following valve delivery, may lead to bleeding or vascular occlusion [22,35,36,37,38]. In a cohort of patients receiving large (>16F) vascular sheaths, VCD failure occurred in 7.6% of the subjects and was predicted by the presence of PVD, a deep skin puncture site and a large body mass index [36]. In a cohort of patients undergoing TAVR with a mean sheath size of 16.8F, VCD failure occurred in 11.4% of the patients and was associated with a smaller femoral artery diameter [37]. In a cohort undergoing TAVR using contemporary 14–16F sheaths, VCD failure occurred in 5.6% of the cases [22]. 

## 4. Percutaneous Management of Vascular Complications

Early detection and rapid management of VC is crucial. Manual compression may achieve hemostasis in cases of minor access site bleeding, and protamine sulfate may be safely administered to reverse anticoagulation [39]. Severe bleeding or limb ischemia may require surgical vascular repair, which may delay patient mobilization and hospital discharge [19]. The goal of performing a minimally-invasive TAVR procedure has provided the impetus for development of novel percutaneous strategies for vascular repair. 

Percutaneous management of VC and vascular repair may be facilitated by placement of a 0.014″ 300-cm length “safety” wire within the femoral artery prior to puncture of the primary vascular access site (e.g., Grand slam, Asahi Intecc, Irvine, CA, USA). The safety wire is delivered via a secondary vascular access site (Figure 1) and is utilized as a rail for delivering devices for diagnosing and treating vascular injury. 

The contralateral femoral artery was traditionally used for secondary vascular access in TF TAVR procedures. Alternative secondary access sites include the ipsilateral distal branches of the femoral artery (superficial femoral artery or profunda) [40,41,42], radial artery [43,44] and brachial artery [45]. Control angiography is performed following sheath removal from the primary vascular access site and VCD deployment. If VC are detected, percutaneous repair may then be performed. Alternatively, an occlusive balloon may be inflated proximal to the site of vascular injury in order to prevent blood loss and stabilize the patient pending open surgical vascular repair.

## 5. Vascular Repair by Stent Graft Implantation

The CFA is subject to repetitive stress and flexion at the hip joint. Implantation of balloon-expandable stents within the CFA is unsafe due to risk of stent fracture, compression, restenosis and branch occlusion [46,47]. Nitinol, a metal alloy of titanium and nickel, has unique shape-memory and super-elasticity characteristics. Nitinol-based devices retain their initial geometry despite undergoing extreme structural deformation. Recent studies have confirmed the safety and efficacy of nitinol stents as an alternative to surgical endarterectomy for revascularizing the CFA in patients with occlusive atherosclerotic vascular disease [48,49,50,51].

Stents grafts (SG), which are covered with impermeable membranes such as polytetrafluoroethylene, may be used for excluding vascular tears and perforations and maintaining arterial integrity. Balloon-expandable SG, which were previously used for endovascular repair of iatrogenic vascular injury [52], have been largely replaced by nitinol self-expanding SG (e.g., Viabahn, Gore Medical, Flagstaff, AZ, USA and Fluency, Bard Medical, Murray Hill, NJ, USA). Several studies have assessed the safety and efficacy of SG implantation for treating vascular access site complications following TF TAVR [53,54,55,56,57,58]. The most common indications for SG implantation were femoral artery rupture and major bleeding (Figure 2); however, SG were also used for treating residual stenosis and significant dissections. 

Stents were implanted in 10–24% of patients undergoing TF TAVR and achieved hemostasis in 98–100%. During follow-up, the stents maintained near-universal patency with a low rate of asymptomatic strut fracture. Anecdotal data suggests that repeat vascular access and deployment of a VCD may be possible within a vessel that was previously treated by SG implantation [59]. A liberal policy of SG implantation has been adopted at some medical centers in order to increase the number of patients undergoing truly percutaneous TF TAVR, including patients with challenging iliofemoral anatomy who are at an increased risk for VC [58].

## 6. Technical Aspects of Stent Graft Implantation

The CFA should be punctured at least 1 cm above the bifurcation in order to enable SG implantation without compromising outflow to the superficial femoral artery or profunda. Decision to implant SG is usually made following sheath removal, when control angiography reveals evidence of VCD failure. 

Long sheaths (e.g., Flexor, Cook Medical, Bloomington, IN, USA) may be used for SG delivery; however, they require upsizing of the secondary vascular access (e.g., delivery of a Fluency SG with a nominal diameter of 8 mm requires a 9F sheath). Alternatively, SG may be delivered by a sheathless technique in order to minimize vascular trauma at the secondary access site. Stent grafts are typically delivered antegradely, from the contralateral femoral artery (“crossover”, “up and over”). Alternatively, ipsilateral distal vascular access within the superficial femoral artery or profunda [40,41,42] and the brachial artery may also be used for SG delivery [45]. The radial artery is unsuitable as a port for SG delivery in most patients due to small vessel diameter and increased distance from the radial artery to the site of vascular injury, which may exceed the length of the SG shaft. Choice of nominal SG diameter should be 1–2 mm larger than the target vessel as measured by pre-procedural CT angiography to ensure apposition of the SG to the vessel wall. 

## 7. Totally Percutaneous TAVR in Cases of Primary VCD Failure

Pre-procedural assessment of the femoral artery anatomy may show that the vascular anatomy is unsuitable for deployment of a VCD. In some patients, attempts to perform pre-closure with a suture-based VCD at the beginning of the TAVR procedure, prior to sheath insertion, may fail. Performing TF TAVR despite the inability to utilize a VCD may necessitate surgical vascular repair of the femoral artery. Patients at increased risk for complications of vascular surgery may benefit from a percutaneous method for achieving access site hemostasis following valve implantation and sheath removal. We assessed the feasibility of a strategy of planned stent graft implantation within the femoral artery for achieving access site hemostasis in a cohort of patients undergoing TF TAVR, in whom vascular pre-closure was not possible (Figure 3) [60].

These patients were considered at increased risk for complications of vascular surgery due to advanced age, frailty, co-morbidities, or immobility. Stent graft implantation achieved access site hemostasis in all patients. During follow-up, 30-day mortality was zero, one-year mortality was 27%, and none of the patients required additional vascular interventions.

## 8. Summary and Conclusions

Recent evolution of techniques and devices for treating severely diseased ileofemoral occlusive vascular disease, as well as novel strategies for utilizing SG for management of access site vascular complications, offer the opportunity to expand the role of totally percutaneous TF TAVR to patients with PVD (Figure 4). 

A systematic approach to procedural planning and management of vascular complications increases the likelihood of procedural success (Figure 5).

Development of TAVR systems with lower device profiles, as well as novel VCD designs with improved efficacy, may expand the role of TF TAVR and decrease the need for alternative vascular access in patients with hostile vascular anatomy.

## Figures and Tables

**Figure 1 jcm-11-02104-f001:**
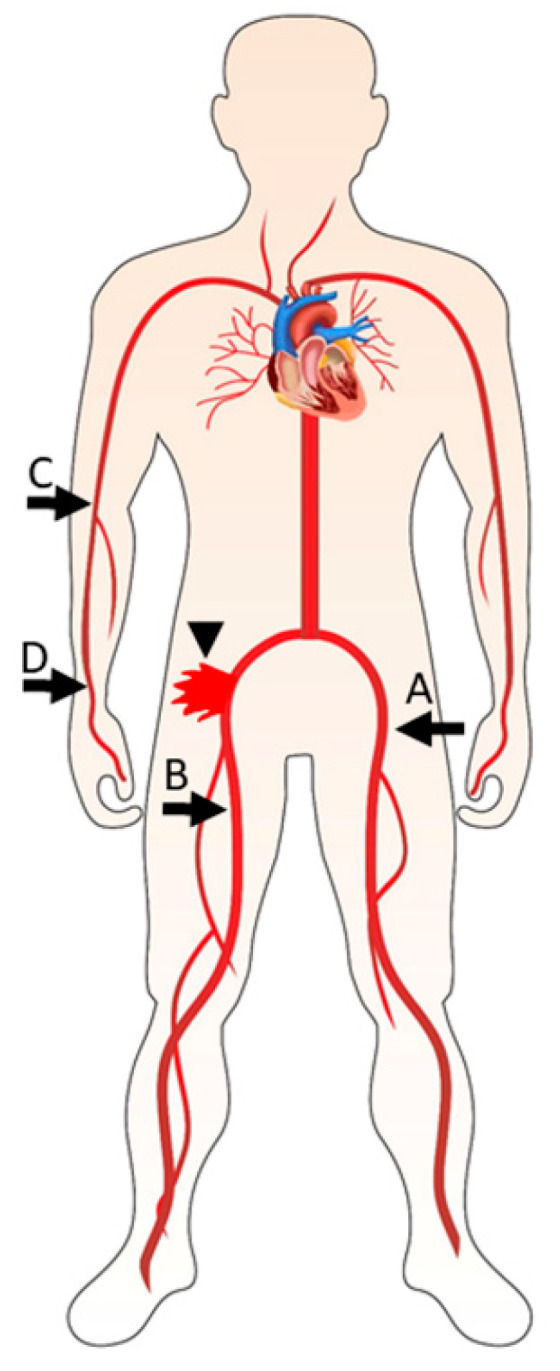
Secondary vascular access sites. Secondary vascular access sites which may be used as ports of entry for a safety wire and delivery of devices for percutaneous repair of femoral artery injury (arrowhead): (**A**) contralateral femoral artery, (**B**) ipsilateral distal femoral artery, (**C**) brachial artery, (**D**) radial artery.

**Figure 2 jcm-11-02104-f002:**
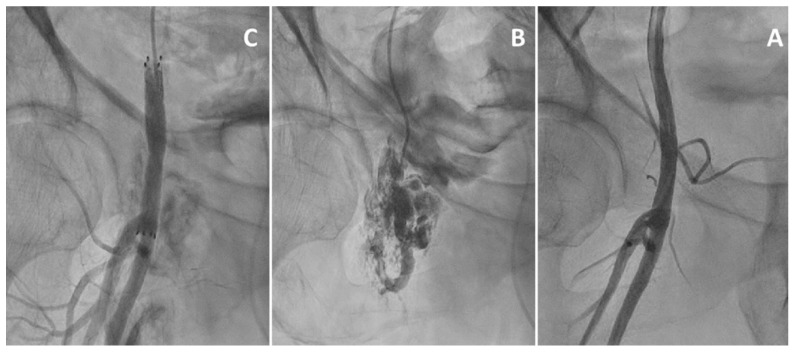
Secondary VCD failure. An 86-year-old female underwent transfemoral TAVR via the right femoral artery (**A**). Vessel pre-closure was performed with a Prostar XL device. Control angiogram performed following removal of the 16 Fr sheath and fastening of the Prostar sutures revealed femoral artery perforation (**B**). A Fluency 9 × 60 mm stent graft which was delivered from the contralateral femoral artery achieved hemostasis (**C**).

**Figure 3 jcm-11-02104-f003:**
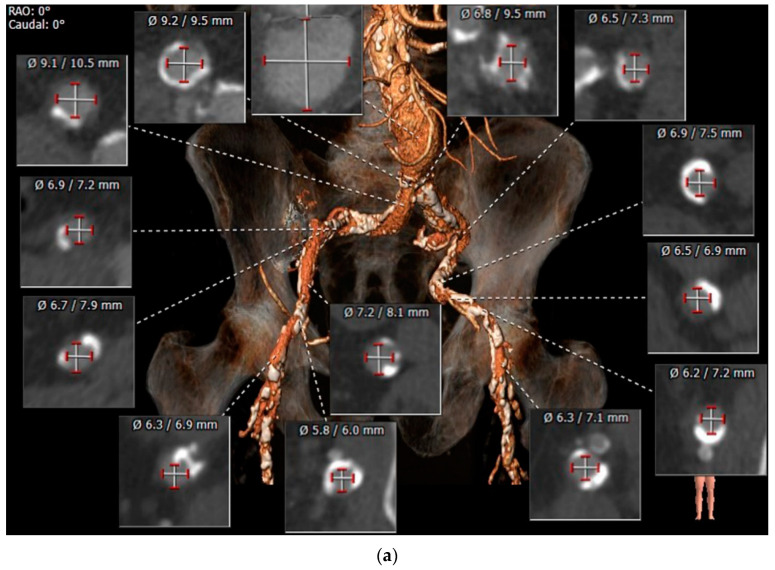
Primary VCD failure. A 90-year-old male with peripheral vascular disease was diagnosed with severe symptomatic aortic stenosis and an abdominal aortic aneurysm. (**a**) CT angiography revealed an infra-renal aortic aneurysm and diffusely calcified ileofemoral arteries with a minimal lumen diameter of 6.0 mm. (**b**) A combined TAVR and endovascular aneurysm repair (EVAR) procedure was performed. 5Fr sheaths were inserted into both superficial femoral arteries, through which two 0.014″ safety wires were delivered to the aorta. Pre-closure with Proglide vascular closure devices was performed in the right femoral artery; however, vascular calcification precluded Proglide deployment in the left femoral artery (**A**). Implantation of a 29 mm Sapien S3 valve (**B**) was followed by EVAR (**C**) (safety wires within the femoral arteries marked with arrows). A 9 × 60 mm Fluency stent graft (between arrowheads) was positioned parallel to the 14Fr sheath (arrow) in the left femoral artery prior to sheath removal (**D**) and deployed following sheath removal (**E**). Proglide devices achieved vascular closure of the right femoral artery following removal of the 18Fr sheath (**F**).

**Figure 4 jcm-11-02104-f004:**
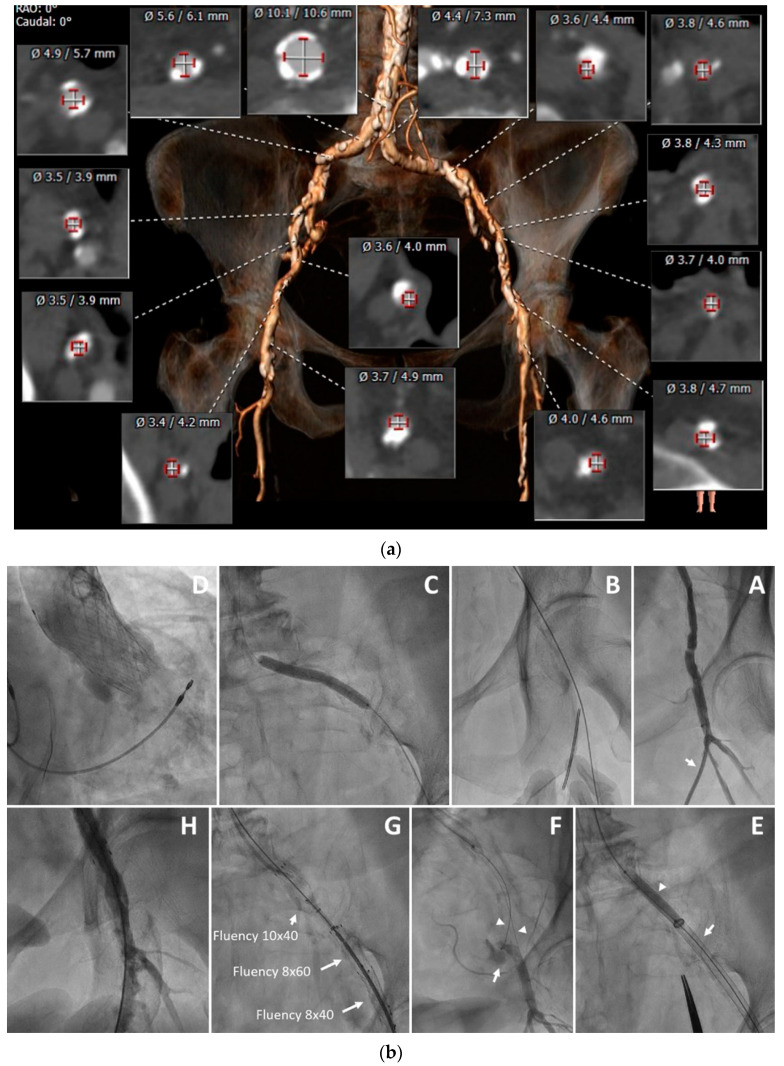
Use of multiple modalities in a patient with hostile vascular anatomy. An 89-year-old female with severe PVD and severe symptomatic aortic stenosis was referred for TAVR. (**a**) CT angiography revealed extremely calcified and stenosed ileofemoral arteries, with minimal diameters of 3.5 mm. (**b**) The TAVR procedure was performed via the left femoral artery, with two safety wires inserted; antegrade via the right brachial artery and retrograde via the left superficial femoral artery (SFA). (**A**) A 5Fr sheath was inserted into the left SFA (arrow). (**B**) The left common femoral artery was then punctured and pre-closure with a Proglide device was performed. (**C**) The left iliac artery was dilated with a 6 mm Shockwave balloon at 6 atmospheres and a 6 mm non-compliant balloon at 24 atmospheres. (**D**) A 26 mm Evolut Pro valve was delivered over a stiff Lunderquist wire and implanted. (**E**) An occlusive 7 mm balloon (arrowhead) was delivered via the right brachial artery and inflated within the proximal left iliac artery prior to sheath removal (arrow). (**F**) Control angiography of the left femoral artery, which was performed via the left SFA following removal of the 16 Fr sheath and fastening of the Proglide sutures, revealed vessel perforation (arrow) and location of both safety wires (arrowheads). (**G**) Three overlapping stent grafts were implanted within the ileofemoral artery. (**H**) Repeat angiography confirmed adequate hemostasis. The SFA and brachial sheaths were removed following administration of protamine sulfate.

**Figure 5 jcm-11-02104-f005:**
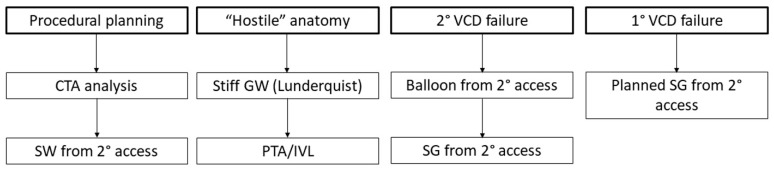
Flowchart for procedural planning and management of vascular complications. CTA, CT angiography; SW, safety wire; GW, guide wire; PTA, peripheral angioplasty; IVL, intravascular lithotripsy; VCD, vascular closure device; SG, stent graft.

**Table 1 jcm-11-02104-t001:** Minimal vessel diameters required for valve delivery.

Valve	Valve Size	Sheath ID	Sheath OD	Vessel MLD	Manufacturer
Evolut R	23/26/29 mm	14F/4.7 mm	6.0 mm	5.0 mm	Medtronic
Evolut R	34 mm	16F/5.3 mm	6.7 mm	5.5 mm	Medtronic
Evolut Pro	23/26/29 mm	16F/5.3 mm	6.7 mm	5.5 mm	Medtronic
Sapien S3	23/26 mm	14F/4.7 mm	6.0 mm	5.5 mm	Edwards
Sapien S3	29 mm	16F/5.3 mm	6.7 mm	6.0 mm	Edwards

ID, internal diameter; OD, external diameter; MLD, minimal lumen diameter.

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
