# Peer review of "Strategies for Facilitating Totally Percutaneous Transfemoral TAVR Procedures"

_jcm, 2022, doi:10.3390/jcm11082104_

Round 1

Reviewer 1 Report

Dear Editor, dear authors, 

with great interest, I have read the article entitled "Strategies for facilitating totally percutaneous transfemoral TAVR Procedures" by Eitan et al.

The article is well written and there is no concern regarding presentation or methodology, neither regarding imaging etc. The most important point from my side is that the metholdology described/reviewed, is at least in part published by the authors just recently [ref. 60]. Although their previous work has been cited adequately, the major concern is the limited novelty of the paper. 

As minor comments, I miss at least the discussion on for example transapical , or transaxillary approaches. Knowing that these access routes are used in distinct patient collectives, such data may not be completely comparable, but when focussing on PVD patients with difficult access routes, these approaches need to be focussed on. Also, considering the relatively high complication rates of TAVR in such complicated patients, the authors should spend some debate on SAVR as well. Notably, of course not all, but many of the patients receiving the TAVR these days can be operated on as well, still with a relatively low complication profile in experienced hands.

The latter aspects are not really deciding acceptance of the manuscript , but discussion on these issues would certainly improve the paper. 

Author Response

Reviewer 1:

  1. With great interest, I have read the article entitled "Strategies for facilitating totally percutaneoustransfemoral TAVR Procedures" by Eitan et al.”: We thank the Reviewer for this comment.
  2. “The article is well written…”: We thank the Reviewer for this comment.
  3. “As minor comments, I miss at least the discussion on for example transapical , or transaxillary approaches. Knowing that these access routes are used in distinct patient collectives, such data may not be completely comparable, but when focusing on PVD patients with difficult access routes, these approaches need to be focused on”: We have added the following text in the Introduction section: “Alternative vascular access approaches (e.g. transapical, transaxillary, transcaval) were developed in order to enable TAVR in patients with hostile vascular anatomy in whom TF valve delivery was considered technically challenging. No randomized trials have compared these different approaches, however the TF route is considered the least invasive. The aim of the present review is to summarize novel techniques which may facilitate totally percutaneous TF valve delivery”.
  4. “Also, considering the relatively high complication rates of TAVR in such complicated patients, the authors should spend some debate on SAVR as well”: We have added the following text in the Introduction section: “Calcified, stenosed and tortuous ileofemoral arteries characterize patients with “hostile” vascular anatomy and may impede valve delivery and increase susceptibility to vascular injury during TAVR. In such patients, SAVR may potentially enable valve replacement without need for instrumentation of the peripheral vasculature. However, patients with peripheral vascular disease often have co-morbidities which are associated with increased surgical risk and may therefore benefit from percutaneous valve delivery”.

Reviewer 2 Report

In this manuscript, Eitan A et al. present a succinct summary of the current trend to totally percutaneous transfemoral TAVR application, discussing its feasibility, associated risks, and management of these risks. The manuscript is, in my opinion, clearly written and well composed, with illustrative figures broadening the reader’s understanding. However, to further clarify the presented data, the authors should consider generating a flow chart figure, delineating a decision tree for handling TAVR complications.

Two additional, very minor comments:

  • Line 13: “…may impede valve delivery, and vascular access…” (i.e. add a comma)-
  • Table 1: While French is a common metric for description of the sheath, in this Table, the authors should also state the ID in mm, as this would make direct comparison to vessel MLD easier.

Author Response

Reviewer 2:

  1. “In this manuscript, Eitan A et al. present a succinct summary of the current trend to totally percutaneous transfemoral TAVR application, discussing its feasibility, associated risks, and management of these risks. The manuscript is, in my opinion, clearly written and well composed, with illustrative figures broadening the reader’s understanding”: We thank the Reviewer for these comments.
  2. “However, to further clarify the presented data, the authors should consider generating a flow chart figure, delineating a decision tree for handling TAVR complications”: We have added the suggested figure (Figure 5).
  3. “Line 13: “…may impede valve delivery, and vascular access…” (i.e. add a comma)-“: We have added the comma as suggested.
  4. “While French is a common metric for description of the sheath, in this Table, the authors should also state the ID in mm, as this would make direct comparison to vessel MLD easier”: we have revised Table 1 to include sheath outer diameter as well as inner diameter and have specified sheath diameters in mm as well as in French size.

Reviewer 3 Report

Thank you for the possibility to review this elegant paper. The topic is necessary and the article is properly written. I have no major comments.

I suggest to order the subsequent Figures according to European practice - from the left to right. 

A citation of a following paper would be beneficial: doi: 10.5603/KP.a2017.0205. 

Anyway, congratulations for this excellent paper. 

Author Response

We thank the Reviewer for these comments.

Round 2

Reviewer 1 Report

Dear Editor, dear authors, 

my comments have been addressed properly. It remains the question regarding general novelty. 

Anyhow, it is comprehensively written.

Author Response

We thank the Reviewer for this comment.